# Mealtime TV Use Is Associated with Higher Discretionary Food Intakes in Young Australian Children: A Two-Year Prospective Study

**DOI:** 10.3390/nu14132606

**Published:** 2022-06-23

**Authors:** Eloise-kate Litterbach, Miaobing Zheng, Karen J. Campbell, Rachel Laws, Alison C. Spence

**Affiliations:** Institute for Physical Activity and Nutrition (IPAN), School of Exercise and Nutrition Sciences, Deakin University, Geelong 3220, Australia; j.zheng@deakin.edu.au (M.Z.); karen.campbell@deakin.edu.au (K.J.C.); r.laws@deakin.edu.au (R.L.); a.spence@deakin.edu.au (A.C.S.)

**Keywords:** family meal, mealtime, socioeconomic, screen use, pre-schooler, diet, fruit, vegetable, discretionary, non-core

## Abstract

Background: Mealtime television use has been cross-sectionally associated with suboptimal diets in children. This study aimed to assess the two-year prospective association between baseline mealtime television use and subsequent diets in young children, and identify socioeconomic differences. Methods: Parents reported their child’s television use at meals, and fruit, vegetable, and discretionary food intakes. Multivariable linear and logistic regression analyses assessed the association between baseline mealtime television use and follow-up diet outcomes. Differences were assessed by socioeconomic position. Results: Participants were 352 Australian parents of children aged six months to six years. Daily mealtime television use (average frequency/day) was associated with higher daily frequency of discretionary food intakes (β 0.2, 95% confidence interval (CI) 0.07–0.67) at the 2-year follow-up. Individually, television use during breakfast and dinner (1–2 days/week compared to never) predicted higher daily intake frequency of discretionary food, β 0.36 (95% CI 0.12–0.60) and β 0.19 (95% CI 0.00–0.39), respectively. Similarly, 3–7 days/week of television use during breakfast and lunch predicted higher frequency of discretionary food intake, β 0.18 (95% CI 0.02–0.37) and β 0.31 (95% CI 0.07–0.55), respectively. Associations were not socioeconomically patterned. Conclusions: Investigating mealtime television use motivators across the socioeconomic spectrum could inform interventions targeting the high consumption of discretionary foods in children.

## 1. Introduction

One of the major contributing factors to poor health outcomes in high-income countries is sub-optimal nutrition, beginning in early childhood. In Australia, children consume well above the recommended intakes for discretionary foods [1,2], well below recommendations for vegetables [3], and an increasing proportion consume insufficient fruit with increasing child age [4]. These behaviours are contributing to high rates of childhood obesity [5] making child nutrition a public health priority. Early childhood is particularly important because nutrition during infancy and the preschool years influences growth and health outcomes across the entire life course [6,7]. During the early years, the majority of young children’s eating occurs within the home [8], and the home environment contributes substantially to children’s diets, eating behaviours, and learned food preferences [9,10]. 

A family’s socioeconomic position (SEP) is also an important correlate of children’s diets and obesity risk [11,12]. Children from disadvantaged backgrounds are more likely to have less healthy diets and poorer health outcomes when compared to those of higher SEP [2,13,14]. In particular, inadequate intakes of fruits and vegetables, and high intakes of discretionary foods, are more commonly seen in children from low socioeconomic groups [2,15].

Meals within the home are a convenient context to promote healthy diets to young children. Regardless of SEP, mealtimes present opportunities to improve children’s dietary outcomes, and the more mealtimes a child shares with their family, the higher the likelihood of them consuming more healthy diets [16,17]. Dinner and breakfast are the focus of most mealtime research [17]. Lunches are less frequently explored [18], and snack times are often overlooked as a ‘mealtime’ [19], or they are considered as a separate concept to other meals [20]. However, including all potential meal types in mealtime research is particularly relevant for young children because of snacking frequency and the large energy and nutrient contribution that snacks make to their diets [21]. 

Television (TV) viewing during mealtimes is an important focus because it is a frequent and potentially modifiable behaviour. Around one third of young children in Australia eat while watching TV at least once every day [19], and this proportion increases as children age [22]. For young children, mealtime TV use appears to be highest during snack time, compared with other meal types [19]. Watching TV during mealtimes is associated with sub-optimal dietary intakes [23]. For example, in young children, mealtime TV use has been associated with higher intakes of sugar-sweetened beverages, and energy-dense, nutrient-poor discretionary foods [24,25]; lower intakes of fruit [26] and vegetables [26,27,28]; and lower overall diet quality [24,29,30]. Two separate systematic reviews assessing mealtime TV use and dietary outcomes have found that meals with the TV on were inversely associated with the quality of children’s diets, from age 1–18 years [20,23]. However, most of this research is cross-sectional [24,26,27,28,29,31,32,33,34,35,36,37,38], prone to reverse causation, and the temporal order of the relationship cannot be established.

Both mealtime TV use [19] and dietary intakes [2] are socioeconomically patterned in children. Therefore, any associations between mealtime TV use and diet may also be socioeconomically patterned. Few studies have considered mealtime TV frequency by SEP, and these suggest an inverse relationship [19,39]. A recent meta-analysis of mealtime components’ influence on the healthfulness of meals suggested that the association between mealtime components (such as TV use during meals) and dietary outcomes hold, regardless of SEP [23]. However, the research focuses on cross-sectional studies only, and longitudinal exploration is required.

Given this, research aiming to understand what influences children’s diets and eating behaviours, and socioeconomic differences, is fundamental. Such research provides a better understanding of how to improve diets and overall health of both children and adults, and, in turn, the health of future generations. Therefore, in this study of Australian children aged six months to six years, we aimed to assess prospective associations between TV use during specific mealtimes (breakfast, lunch, dinner, and snacks) and fruit, vegetable, and discretionary food intakes at the subsequent two-year follow-up, and determine if associations were socioeconomically patterned. 

## 2. Materials and Methods

### 2.1. Study Participants

This study used data from the Family Meals with Young Kids Study, an online survey completed in 2014 [19], and followed up two years later. At baseline, parents or carers with children aged between six months and six years, able to speak and read English and living in Australia, were eligible to participate. The study was approved by Deakin University (HEAG-H 55_2014).

Recruitment was conducted online using social media to advertise the study on blogs, Facebook™, and online parent groups. Participants followed a link to an online screening tool and survey, hosted on SurveyMonkey™ [40]. Follow-up was conducted two years later by recontacting participants via email and inviting them to complete a follow-up survey.

### 2.2. Mealtime TV Use

At baseline, mealtime TV use was assessed by asking ‘how often does (child’s name) watch television while eating (each of breakfast/lunch/dinner/snacks)?’ [41]. Five response options were available: never or <1 day per week, 1–2 days/week, 3–4 days/week, 5–6 days/week, and 7 days/week. These items and the response scale were informed by previous research [42,43]. Given the data distribution, where there was a high proportion of ‘never’ responses (38–66%), the five response options of mealtime TV use were collapsed into three categories (never, 1–2 days/week, and 3–7 days/week) for subsequent analyses. 

To calculate average daily mealtime TV use, as conducted previously [19], a cumulative measure was first created out of a possible 28 occasions per week (four possible occasions per day: breakfast, lunch, dinner, and snack time). Where the response option was a range, the middle value was imputed in the cumulative measure (for example, 1–2 days was converted to 1.5 days). The weekly score out of 28 was divided by seven to create a daily average score out of four, which was treated as a continuous variable for analysis. The survey also assessed the use of other electronic media during mealtimes, using a separate question with the same format as TV use. However, very few (2–8%) children engaged in other electronic media at each mealtime; therefore, there was insufficient variability in the data to include in analysis (results not presented). 

### 2.3. Intakes of Discretionary Foods, Fruits, and Vegetables

All dietary measures have been previously validated [44,45,46] and were considered the most relevant for this study at the time of data collection [47]. At both baseline and follow-up, we asked ‘In the past month, about how often has (your child) had the following foods?’. Eight categories covering a range of discretionary items were included, with examples specific to Australia: fried potato products, savoury biscuits, sweet biscuits, takeaway food, baked goods, confectionary, and sugar-sweetened beverages [44]. Six response options were available: never or <once/month, 1–3 times/month, once/week, 2–4 times/week, 5–6 times/week, and 7 times/week. Intakes of discretionary foods and beverages are henceforth referred to as ‘discretionary foods’. Appendix B includes a list of discretionary food items assessed. Intake frequency of each discretionary food item over the past month was converted to daily intakes using daily equivalent frequencies (never = 0; 1–3/month = 0.067; 1/week = 0.143; 2–4/week = 0.429; 5–6/week = 0.786; 1/day = 1.0; 2–3/day = 2.5; 4–5/day = 4.5; ≥6/day = 6.0) [48]. The daily frequency of each item was then summed to create average daily discretionary food frequency, analysed as a continuous variable (frequency per day).

Fruit and vegetable intakes were assessed by one item each, asking about ‘how many serves of vegetables does (child’s name) usually eat per day?’ (with examples), and included five response options: my child does not eat these, <1 serve/day, 1 serve/day, 2 serves/day, 3 serves or more/day. Informed by the data distributions, intakes of fruit and vegetable were dichotomised for analysis as <2 and ≥2 serves per day, respectively. 

### 2.4. Covariates

The baseline questionnaire also collected child age, child gender, location of meals, and family meal frequency. These variables have been considered as potentially influencing the association between diet and shared mealtimes [17,49], and were, therefore, included as covariates. Data collection methods for these measures have been described elsewhere [19]. The location of meals was dichotomised into an optimal category: sitting at a table/bench (highchair or chair), and a sub-optimal category: in the car, moving around the house, and sitting on the floor/couch. Family mealtime frequency for each meal type was defined as the responding parent and child eating together, and was categorised into ‘never’, occasionally (1–2 days), and more frequently (3–7 days) for the current analysis.

### 2.5. Socioeconomic Position

The education level of the responding parent at baseline was used to define SEP for this study. Previous research has found maternal education to be the strongest and most consistent socioeconomic predictor of children’s dietary intakes [2,11,39,50], and the majority of responding parents (97%) in this study were mothers. Therefore, parental education was used as a proxy for SEP. 

Due to the large number of responding parents who were university educated in this sample (>70%), education was dichotomised to ‘university educated’ and ‘non-university educated’; comprising ≤ 12 year or equivalent, trade/apprenticeship (e.g., hairdresser, chef), and certificate/diploma (e.g., childcare, technician).

### 2.6. Survey Reliability

A test–retest study was conducted at baseline to measure the reliability of survey questions. A subsample of 54 participants completed the same survey again, two weeks after initial completion. Intraclass correlations (ICCs) were performed for each ordered categorical variable, and Cohen’s Kappa was performed for each non-ordered categorical variable. 

Intraclass correlations assessing the reliability of dietary measures ranged from moderate (ICC 0.5) to good (ICC 0.81). The reliability of the mealtime TV use measure ranged from good (ICC 0.77) to excellent (ICC 0.96) (Table 1). The interpretation of values was based on Koo and Li’s criteria of moderate (ICC = 0.5–0.75), good (ICC = 0.75–0.9), and excellent reliability (ICC > 0.9) [51].

### 2.7. Statistical Analysis

Descriptive analyses were conducted to explore the sample characteristics. Multivariable logistic or linear regressions were used to examine the prospective associations between mealtime TV use at baseline and intakes of fruit and vegetables (<2 and ≥2 serves per day), and discretionary foods (intake frequency per day), at the 2-year follow up, respectively. Separate models were conducted for each meal type (breakfast, lunch, dinner, and snacks), as well as average daily mealtime TV use. Crude models were conducted first with mealtime TV use as the exposure, and dietary intakes at follow-up as the outcome with adjustment for baseline dietary intakes. 

Adjusted models included the following covariates: child age, child gender, family mealtime frequency, SEP, and mealtime location. Pearson and Spearman correlations were conducted to test for potential multicollinearity between mealtime TV use and covariates. No significant correlation was found, so all covariates were included simultaneously in the adjusted model. Analyses to detect possible interaction between mealtime TV use and covariates were also performed by creating interaction terms (the product of two variables). The interaction between mealtime TV use and mealtime location was significantly associated with dietary intake; therefore, that interaction term was also included as a covariate in the adjusted model. 

To test if the association between mealtime TV use at baseline and dietary intakes at follow-up differed by SEP, we conducted stratified analyses (crude and adjusted) by SEP, and the likelihood ratio test was used to test for differences between stratum-specific regression coefficients. For adjusted analysis, the model adjusted for all aforementioned covariates, except for SEP. All analyses were conducted in StataIC (version 1.0, StataCorp., College Station, TX, USA) with statistical significance set at *p* < 0.05. 

## 3. Results

### 3.1. Participant Demographics

Of 992 participants at baseline, 369 participants with complete data on mealtime TV use at baseline and dietary intakes at both baseline and follow-up were included in the crude model. Seventeen participants had missing data on covariates, resulting in a final sample of 352 included in the adjusted model (Figure 1). A comparison of demographics between respondents and non-respondents at follow-up showed no significant differences between the groups (Appendix A Table A1).

Table 1 summarises the baseline characteristics of participants. A large proportion of respondents were university educated (72%). Child age distribution at baseline was 31% aged 6 months to <1.5 years, 29% aged 1.5–<3 years, and 40% aged 3–<6 years, with an almost equal number of males and females. 

At baseline, most children (74–86%) had optimal eating locations for main meals (breakfast, lunch, and dinner). In contrast, most children (71%) had suboptimal eating locations for snacks. A large proportion of parents ate each meal type with their children 3–7 days per week (74–90%). Around one third of children ate breakfast, lunch, and/or dinner while watching TV at least once per week, and a similar number ate snacks while watching TV on 3–7 occasions per week.

### 3.2. Dietary Outcomes

At baseline, 71% and 54% of children consumed ≥ 2 serves of fruits and ≥2 serves vegetables, respectively. At follow-up, 82% of children consumed ≥ 2 serves of fruits, and 69% consumed ≥ 2 serves of vegetables. The mean daily frequency of discretionary foods was 0.96 (standard deviation (SD) 0.88) at baseline, and 1.30 (SD 0.85) at follow-up.

### 3.3. Baseline Television Use and Prospective Dietary Outcomes

Crude analysis revealed that overall daily mealtime TV use at baseline was associated with a higher frequency of consuming discretionary foods at 2-years follow-up (β: 0.16; *p* < 0.01) (Table 2). No evidence of association was found for total daily mealtime TV use at baseline and prospective intakes of fruit or vegetables. Mealtime TV use (≥1 day per week versus never) during breakfast was associated with a higher frequency of consuming discretionary foods (*p* = 0.03). Significant associations were also found between mealtime TV use during lunch (3–7 days per week), dinner (1–2 days per week), and snack times (3–7 days per week), and a higher frequency of discretionary foods. No significant associations were found between baseline mealtime TV use and vegetable intakes at follow-up, although dinner TV use 3–7 times per week showed a trend towards association with lower odds of consuming ≥ 2 serves of vegetables/day (odds ratio (OR): 0.56; *p* = 0.07). For fruit, mealtime TV use at lunch (1–2 days/week) and dinner (1–2 days/week) was associated with lower odds of consuming ≥ 2 serves of fruit per day, with odds ratios of 0.47 (*p* = 0.04) and 0.48 (*p* = 0.03), respectively.

In the adjusted model (Table 3), the significant association between baseline mealtime TV use and discretionary food frequency at follow-up remained for most meals, except snacks. The significant mealtime TV predictors of fruit intake observed in the crude model were attenuated, and results related to follow-up vegetable intakes were similar.

### 3.4. Socioeconomic Position

In the multivariate analyses, parent education was not significantly associated with baseline mealtime TV use at any of the assessed mealtimes and prospective intakes of fruit, vegetables, and discretionary foods (not reported in tables; *p* = 0.15–0.82). Stratified analyses by two SEP groups are presented in Appendix A Table A2. There were no significant associations observed between mealtime TV use and diet for children whose mothers were non-university-educated. In children whose mothers were university-educated, higher mealtime TV use frequency 1–2 days per week for breakfast (β: 0.39; *p* = 0.01) and dinner (β: 0.24; *p* = 0.05), and 3–7 days per week for breakfast (β: 0.23; *p* = 0.03) and lunch (β: 0.39; *p* = 0.01), were significantly associated with discretionary food frequency. Additionally, watching TV during dinner 1–2 days per week was associated with lower odds of consuming ≥ 2 serves of fruit per day in this group (OR 0.41; *p* = 0.05). In both SEP groups, mealtime TV use was not associated with vegetable intakes (*p* < 0.05). 

## 4. Discussion

In a cohort of Australian children between 6 months and 6 years, a higher frequency of mealtime TV use was prospectively associated with higher discretionary food intakes at the 2-year follow up. These results were broadly consistent for breakfast, lunch, and dinner times, as well as total mealtime TV use across the day. 

Previous cross-sectional research has shown a consistent association between young children’s mealtime TV use and sub-optimal diets, particularly lower intakes of fruit and vegetables, and higher discretionary food intakes [20,24,26,27,28,29,31,32,33,34,35,36,37,38]. The present study is the first to evaluate the prospective association between mealtime TV use and dietary intake in young children, and thus, provides insights on the direction of influence. The significant prospective association between higher mealtime TV use and higher frequency of discretionary food intakes observed emphasises the importance of understanding the mechanisms by which this association plays out. Previous literature has highlighted that in adults, TV use during meals can attenuate feelings of fullness, leading to a higher consumption of food in one eating occasion, and contributing to an increased overall energy consumption [52]. In addition, regularly eating while watching TV may, in turn, result in TV triggering snacking, even when not hungry [53]. Further, watching TV is highly likely to expose children and families to discretionary food (‘junk food’) advertising [29], which is known to influence targeted food consumption. There are currently no regulations on discretionary food advertising in Australia. These factors, and others, might represent a cluster of sub-optimal eating behaviours among families more likely to engage in mealtime TV use.

Watching TV during breakfast showed the strongest prospective associations with discretionary food consumption. This is interesting because sugary breakfast cereals were not included in our list of discretionary items assessed, highlighting some important considerations. Among these is whether the association between TV use at breakfast and discretionary food intake stems from the food eaten at breakfast, or could it be that a clustering of less than optimal dietary and mealtime TV behaviours might play a role in determining dietary intake? For example, families who engage in breakfast TV use might also tend to purchase and consume more discretionary foods throughout the day. Acknowledging clustering in mealtime research is important [16,23]. Regardless, the prospective nature of this study suggests that mealtime TV use, particularly at breakfast, may be a determinant of diet outcomes in children.

Interestingly, adding location as a covariate reduced the strength of associations between TV use during snack times and the consumption of discretionary foods, fruits, and vegetables (not reported in tables), and the association with discretionary food intakes was no longer significant. Eating in a location other than at a table/bench (such as in the lounge room or the bedroom) has been found to be cross-sectionally associated with reduced vegetable intake [54], poorer overall diets in young children [55], and higher weight status in slightly older children [27]. Little research has focussed on younger children’s mealtime location. It is possible that the consumption location may influence the types of foods parents offer their children. For example, one possibility could be that parents consider that meals eaten on the couch require foods which are easier to hold, can be eaten with fewer utensils, or that produce less mess. Alternatively, parents might intentionally offer less-preferred foods (such as vegetables) while children are distracted by screens in an attempt to increase their intake. Furthermore, families of lower SEP may have fewer resources which enable them to eat in optimal locations, such as space for, or ownership of, a dining table [55], as well as the possible clustering of sub-optimal mealtime behaviours in some families [56]. Understanding potential interactions between mealtime TV use, location, and discretionary food intakes and linked contributors requires exploration. In particular, considering strategies which might address the behaviours in combination and whether these strategies are feasible and acceptable for parents of young children to implement is crucial.

This study highlights the importance of understanding young children’s TV use during snacking, in the context of mealtimes. Given that snacks contribute substantially to young children’s overall diet intake and diet quality [57], but are little studied in the context of mealtimes [17,20], the inclusion of snacks in this research is an important contribution. In this study, snacks were found to be least frequently eaten at a table or bench, compared to other meal types. Additionally, young children’s mealtime TV use appears to be the highest during snack time [19], as well as the meal least-shared with parents [19]. This highlights that snack times may offer the greatest opportunity for improvements in young children’s mealtime behaviours. 

Although this study found no significant associations between mealtime TV use and prospective intakes of fruit and vegetables, all results were in the expected direction, with intakes lower as TV use increased. Previous cross-sectional studies supporting this association have used measures such as four-day [27] or seven-day dietary recall [26,37], TV time diaries [35], and Ecological Momentary Assessments [28], all of which are more detailed than measures in the present study and may have enabled the detection of smaller differences. It is noteworthy that in this study, there was minimal change in fruit and vegetable intakes from baseline to follow-up, which is consistent with existing evidence that young children’s fruit and vegetable intakes tend to change little between the ages of nine months and five years [2]. This could contribute to explaining the minimal associations found between mealtime TV use and fruit and vegetable intakes in the present study. Alternatively, the prospective association between mealtime TV use and diet may well be more closely aligned with discretionary intakes than that of fruits and vegetables. This may provide a rationale for focusing on mealtime TV behaviours in nutrition promotion interventions which focus on limiting discretionary food intakes in young children.

This study did not find a differential effect of SEP on associations between mealtime TV use and diet. The strengths of the associations are similar, suggestive of the stratified analysis by SEP groups resulting in the low SEP group being too small to detect true differences. The results suggest that mealtime TV use is prospectively associated with discretionary food intakes, irrespective of SEP. This supports some cross-sectional research that mealtime components, such as TV, may not be influenced by SEP [23]. Although this study found no statistical differences, other evidence suggests that low SEP families are more likely to watch TV during meals [19,36,58], and, therefore, may be more susceptible to poorer dietary outcomes. The exploration of mealtime TV use in specific socioeconomic groups will help to inform targeted nutrition promotion initiatives.

To our knowledge, this is the first study to assess the prospective association between mealtime TV and intakes of fruit, vegetables, and discretionary foods. Given that the majority of mealtime research is cross-sectional, providing clarity of the temporal order of the outcomes measured is an important strength of this study. It is also the first study to assess a range of eating occasions across the day, including snacks. A potential limitation of this study may be the dietary measures, given their brevity, and that in the current study, child fruit and vegetable intakes were reportedly higher than the Australian average [3]. Though this may reflect variations in the dietary measures used between studies, it is possible that it reflects that higher child dietary quality is associated with a high level of parental education [59]. In the current study, 73% of parents were university educated versus ≈ 49% of Australian women aged 25–34, from national data [60], highlighting a need for future research focussing on populations of lower education. Moreover, although validated tools were used for intakes of all dietary items, future analysis would benefit from using a more detailed and robust measure of dietary intake. Additionally, snacks were conservatively included only once per day in calculating daily mealtime TV use. Future research could assess the proportion of children’s meals eaten while using screens, accounting for how many meals an individual child eats in a day. Other limitations to this study include the potential influence of other unmeasured confounders, such as child ethnicity, body mass index (BMI), and other dietary outcomes, as well as potential residual confounding.

## 5. Conclusions

This study identified that a higher frequency of mealtime TV use in young children was prospectively associated with a higher frequency of discretionary food intakes two years later, irrespective of SEP. Reducing mealtime TV use could be a promising intervention strategy in the pursuit of reducing young children’s discretionary food intakes, the promotion of optimal diet quality, and improving health outcomes for young children and future generations.

## Figures and Tables

**Figure 1 nutrients-14-02606-f001:**
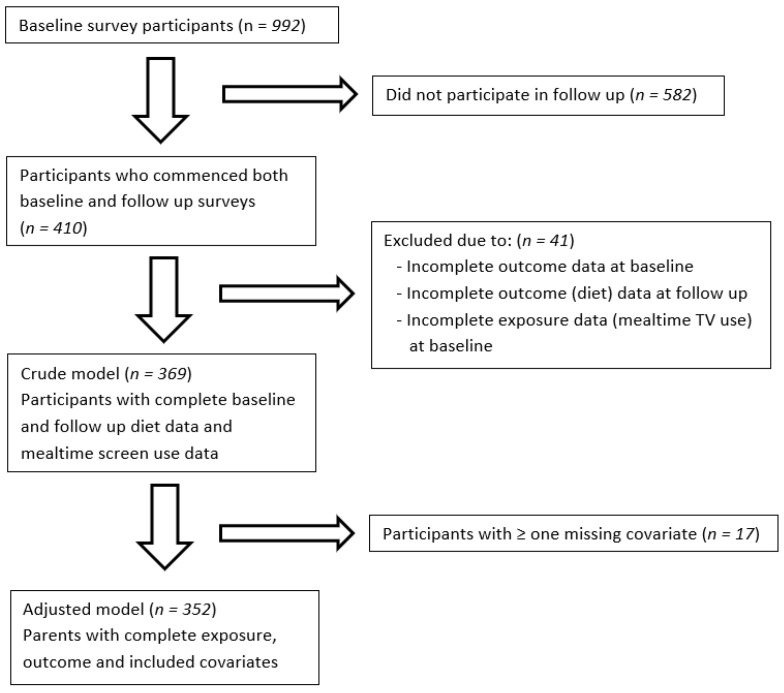
Participant study inclusion criteria. Covariates in adjusted models include location of meals, family meal frequency, parent education, and child age and gender. TV: Television.

**Table 1 nutrients-14-02606-t001:** Participant characteristics (*n* = 369).

		*n* (%)
Parent education	Below university equivalent	97 (26)
	University degree	272 (74)
Child gender	Male	192 (52)
	Female	177 (48)
Child age (years) mean (SD)	Baseline	2.5 (1)
	Follow-up	4.3 (1)
Eating location (baseline)		
Breakfast (*n* = 359)	Sub-optimal	83 (23)
	Optimal	276 (77)
Lunch (*n* = 304)	Sub-optimal	80 (26)
	Optimal	224 (74)
Dinner (*n* = 365)	Sub-optimal	53 (15)
	Optimal	312 (86)
Snacks (*n* = 317)	Sub-optimal	226 (71)
	Optimal	91 (29)
Family meal frequency (baseline)		
Breakfast (*n* = 368)	never	22 (6)
	1–2 days	46 (13)
	3–7 days	300 (82)
Lunch (*n* = 365)	never	15 (4)
	1–2 days	58 (16)
	3–7 days	292 (80)
Dinner (*n* = 365)	never	19 (5)
	1–2 days	26 (7)
	3–7 days	320 (88)
Snacks (*n* = 361)	never	28 (8)
	1–2 days	69 (19)
	3–7 days	264 (73)
Mealtime TV use frequency (baseline)		
Breakfast (ICC 0.96)	never	234 (63)
	1–2 days	38 (10)
	3–7 days	97 (26)
Lunch (ICC 0.88)	never	254 (69)
	1–2 days	57 (16)
	3–7 days	58 (16)
Dinner (ICC 0.77)	never	246 (67)
	1–2 days	62 (17)
	3–7 days	61 (17)
Snacks (ICC 0.77)	never	154 (42)
	1–2 days	88 (24)
	3–7 days	127 (34)
Summed daily frequency	mean = 0.7 SD = 0.8	range 1–4

Eating location optimal category: sitting at a table/bench; Sub-optimal category: in the car, moving around the house, and sitting on the couch; Family meal definition: frequency of child eating a meal with responding parent; ICC: Intraclass correlations; SD: Standard deviation.

**Table 2 nutrients-14-02606-t002:** Crude model. Association between children’s mealtime TV use at baseline and consuming two or more serves of fruit and vegetables per day, and consuming non-core discretionary foods on more than one occasion per day at the follow-up, adjusting for baseline intakes.

	Discretionary Food		Daily Vegetable Intakes		Daily Fruit Intakes	
	(Daily Frequency)		(≥2 Serves/Day)		(≥2 Serves/Day)	
	β (95% CI)	*p* Value	OR (95% CI)	*p* Value	OR (95% CI)	*p* Value
Daily mealtime TV use (frequency/day)	0.16 (0.07–0.25) ^a^	**<0.01**	0.87 (0.66–1.14) ^b^	0.32	0.79 (0.58–1.10) ^b^	0.11
Breakfast TV use						
<1 day per week	(REF)					
1–2 days per week	0.27 (0.03–0.51)	**0.03**	1.04 (0.47–2.30)	0.91	0.74 (0.30–1.86)	0.52
3–7 days per week	0.19 (0.01–0.36)	**0.03**	0.87 (.51–1.49)	0.62	0.58 (0.32–1.06)	0.08
Lunch TV use						
1–2 days per week	0.09 (−0.11–0.30)	0.36	0.79 (0.41–1.50)	0.46	0.67 (0.33–1.37)	0.27
3–7 days per week	0.30 (0.10–0.51)	**<0.01**	0.79 (0.41–1.50)	0.46	0.47 (0.24–0.95)	**0.04**
Dinner TV use						
1–2 days per week	0.20 (0.00–0.39)	**0.05**	0.66 (0.36–1.25)	0.20	0.48 (0.24–0.95)	**0.03**
3–7 days per week	0.15 (−0.05–0.35)	0.13	0.56 (0.30–1.04)	0.07	0.71 (0.34–1.50)	0.37
Snack time TV use						
1–2 days per week	0.00 (−0.19–0.19)	1.00	0.81 (0.44–1.49)	0.50	0.66 (0.32–1.37)	0.27
3–7 days per week	0.21 (0.03–0.39)	**0.02**	0.84 (0.49–1.44)	0.53	0.57 (0.30–1.06)	0.08

β: coefficient; 95% CI: 95% Confidence Interval; OR: Odds Ratio; REF: Reference; TV: Television; ^a^ denotes frequency/day, ^b^ denotes serves/day. Numbers in bold denote significant outcomes.

**Table 3 nutrients-14-02606-t003:** Adjusted models. Association between children’s mealtime TV use at baseline and consuming two or more serves of fruit and vegetables per day, and consuming non-core discretionary foods on more than one occasion per day at the follow-up, adjusting for baseline intakes and covariates (footnote).

	Discretionary Food		Daily Vegetable Intakes		Daily Fruit Intakes	
	(Daily Frequency)		(≥2 Serves/Day)		(≥2 Serves/Day)	
	β (95% CI)	*p* Value	OR (95% CI)	*p* Value	OR (95% CI)	*p* Value
Daily mealtime TV use (frequency/day)	0.16 (0.07–0.67) ^a^	**<0.01**	0.84 (0.62–1.13) ^b^	0.24	0.85 (0.61–1.8) ^b^	0.33
Breakfast TV use						
≤1 day per week	(REF)					
1–2 days per week	0.36 (0.12–0.60)	**<0.01**	0.92 (0.39–2.19)	0.86	0.96 (0.35–2.58)	0.93
3–7 days per week	0.18 (0.02–0.37)	**0.03**	1.00 (0.55–1.84)	0.99	0.73 (0.37–1.44)	0.37
Lunch TV use						
1–2 days per week	0.01 (−0.22–0.24)	0.90	0.77 (0.34–1.71)	0.52	0.85 (0.35–2.03)	0.71
3–7 days per week	0.31 (0.07–0.55)	**0.01**	0.73 (0.33–1.63)	0.45	0.46 (0.20–1.08)	0.08
Dinner TV use						
1–2 days per week	0.19 (0.00–0.39)	**0.05**	0.68 (0.35–1.33)	0.26	0.58 (0.28–1.21)	0.15
3–7 days per week	0.15 (−0.05–0.35)	0.13	0.56 (0.29–1.10)	0.09	0.84 (0.37–1.90)	0.66
Snacks TV use						
1–2 days per week	0.00 (−0.22–0.21)	0.98	0.78 (0.36–1.70)	0.54	0.85 (0.37–1.97)	0.71
3–7 days per week	0.16 (−0.04–0.36)	0.13	0.66 (0.36–1.30)	0.22	0.75 (0.36–1.59)	0.45

OR: Odds Ratio; β: Coefficient; 95% CI: 95% Confidence Interval; ^a^ denotes frequency/day; ^b^ denotes serves/day. Covariates: child gender, child age, SEP, family meal frequency, location of meals, and interaction term between mealtime TV use and location of meals. Numbers in bold denote significant outcomes.

## Data Availability

The data presented in this study are available on request from the corresponding author. The data are currently not publicly available.

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
