# Peer review of "Mealtime TV Use Is Associated with Higher Discretionary Food Intakes in Young Australian Children: A Two-Year Prospective Study"

_nutrients, 2022, doi:10.3390/nu14132606_

Round 1

Reviewer 1 Report

In this study, Eloise-Kate litterbach et al. discussed the relationship between mealtime TV use and children's food intake. The authors concluded that higher frequency of mealtime TV use in young children was prospectively associated with higher frequency of discretionary food intakes two years. Reducing mealtime TV use may be a promising intervention strategy to reduce children's discretionary food intakes. Children's health is an issue that every country attaches great importance to, so this research on children's nutrition is particularly important.

However, there are several important issues that may require further analysis and interpretation by the authors.

  1. We did not see the distribution of children's age in this study, the authors only mentioned the mean and standard deviation of age, and we did not see the distribution of children's food intake by age group. As far as I know, in most regions, children develop rapidly, and there is a large variation in dietary preferences and types of diets among children of different ages. For example, the cognition and preferences of 2-year-old children and 6-year-old children on discretionary foods (fried potato products, savoury biscuits, sweet biscuits, take away food, baked goods, confirmatory and sugar sweetened beverages) are obviously different. And for most young children (0-3 years old), parents may not allow them to eat this garbage Food, (because many foods are forbidden for children under the age of 3), so I think this may be a bias in this study. The authors require further analysis to fully clarify this point.
  2. We have noticed that not all children eat breakfast every day. Therefore, for those who do not eat breakfast, the number of mealtime TV use will naturally decrease. For example, children who eat breakfast only once a week and watch TV once a week have different effects on the model than children who eat breakfast seven times a week but only watch TV once a week. Therefore, I suggest that this bias may be avoided by stratified analysis of mealtime TV use and other various factors, or by converting the number of mealtime TV use to percentage format (TV watching times / meal times).
  3. Race and children's baseline BMI and other nutritional status are not mentioned in this manuscript, because children with different characteristics may have different preferences and cognition. More demographic adjustment and sensitivity analysis are encouraged, which will make the research more sufficient.

My other comments to improve this manuscript are listed below:

  1. Please pay attention to the format of Apendix tableB;
  2. Please explain whether there is multicollinearity between independent variables;
  3. How did the authors check if the assumptions of Pearson's correlation test (such as roughly normal distribution of variables or lack of outliers) was satisfied?
  4. Mealtime TV use may be related to many factors. Interactive analysis of all factors can be carried out, and the corresponding results can be listed in the appendix;
  5. In the selection of research population, the author selected online recruitment. Can the people selected in this way represent all the people? You know, a lot of people with low education level who probably don't engage much in social media (Twitter, Facebook, etc.).
  6. When searching the literature in this research direction, we found many similar studies such as “Assessing eating context and fruit and vegetable consumption in children: new methods using food diaries in the UK National Diet and Nutrition Survey Rolling Programme”published in 2012 and a systematic review published in 2016, “Associations between children’s diet quality and watching television during meal or snack consumption: A systematic review”. Although this study is a prospective study, please fully explain the innovation  and limitations of this study.

Reviewer 2 Report

Dear Authors

I’m glad to have the opportunity to review your manuscript. You presented interesting research on the high-tech environment and its influence on habits and health.

The manuscript is well written, but I noticed some flawns needed explanation or modification as you see pointed below.

The title has to be modified. Its present form seems to be unattractive to readers. 
The methodology should clearly state it is a prospective study. This information is explained in the latter part of the manuscript.

The population is very scarce, and extrapolation of this data to the wider population is inappropriate. Please explain why/how this group is representative. 
High educated representatives are the main part of this group. Does it correlate with education level in Australia?
Another important lack in this study is information lack regarding the health/behavior development of the studied group in evaluation period.

Author Response

Thank you for your review of this manuscript. Please see the attachment for responses.

Round 2

Reviewer 1 Report

I have read the author's response and manuscript in detail, and believe that although there are still some limitations in the research, the author has explained these problems in detail and reached the level that can be published.

Reviewer 2 Report

Dear Authors

Thank you for your response. Manuscript in present form can be publish